# The Timing and Intensity of Social Distancing to Flatten the COVID-19 Curve: The Case of Spain

**DOI:** 10.3390/ijerph17197283

**Published:** 2020-10-06

**Authors:** Miguel Casares, Hashmat Khan

**Affiliations:** 1Departamento de Economía and INARBE, Universidad Pública de Navarra, Campus de ArrosadÍa, 31006 Pamplona, Spain; 2Department of Economics, Carleton University, C-870 Loeb Building, 1125 Colonel By Driver, Ottawa, ON K1S 5B6, Canada; hashmat.khan@carleton.ca

**Keywords:** COVID-19 pandemic, calibrated model simulations, social distancing, intensity, timing, policy design

## Abstract

The continued spread of COVID-19 suggests a significant possibility of reimposing the lockdowns and stricter social distancing similar to the early phase of pandemic control. We present a dynamic model to quantify the impact of isolation for the contagion curves. The model is calibrated to the COVID-19 outbreak in Spain to study the effects of the isolation enforcement following the declaration of the state of alarm (14 March 2020). The simulations indicate that both the timing and the intensity of the isolation enforcement are crucial for the COVID-19 spread. For example, a 4-day earlier intervention for social distancing would have reduced the number of COVID-19 infected people by 67%. The model also informs us that the isolation enforcement does not delay the peak day of the epidemic but slows down its end. When relaxing social distancing, a reduction of the contagion probability (with the generalization of preventive actions, such as face mask wearing and hands sanitizing) is needed to overcome the effect of a rise in the number of interpersonal encounters. We report a threshold level for the contagion pace to avoid a second COVID-19 outbreak in Spain.

## 1. Introduction

On 11th March 2020, the World Health Organization declared the Coronavirus Disease 2019 (COVID-19) outbreak a pandemic—a worldwide spread of the disease. As of 24th September, there are 978,284 reported deaths due to COVID-19 worldwide, and the total number of confirmed cases has reached nearly 32 million. Unfortunately, the pandemic is still in progress, unleashing a global health crisis and putting enormous pressure on health care systems. The travel-related source of virus spread was quickly followed by “community spread” where the initial source of the infection remains unidentified. During the early phase of the pandemic, governments and public authorities implemented mandatory actions to contain the virus’s spread, such as travel restrictions; lockdowns; closures of public spaces, institutions, and businesses; social (and physical) distancing; and self-isolation. In many countries, these measures reduced the first wave of COVID-19, and together with increased testing and tracking and improved understanding of the airborne-virus spread, allowed a phased reopening of the economy. However, the rising cases, primarily in urban areas, in the United States, Brazil, and India, show that there remains a significant chance of re-imposing lockdowns. Against this backdrop, it remains important to understand how the timing (when to impose the lockdown) and intensity (how strict in terms of actual physical contacts) impact the spread of COVID-19. Our paper contributes to enhancing knowledge on these policy-relevant topics.

Drawing on the epidemiological susceptible–infected–recovered (SIR) methodology, pioneered by [1], we present a discrete-time dynamic model to predict the COVID-19 contagion. Even though the model is simple, it captures the main characteristics of the contagion process and provides insights valuable for policy orientation. We calibrated the model’s parameters to aggregate Spanish data and present simulations to show the dramatic implications of enforcing mobility constraints over the COVID-19 spread in Spain. We also present three scenarios that may occur as social distancing actions are eased, with a possible second peak in the contagion curve.

Our paper can be connected to several recent contributions. Reference [2] conducted a similar exercise to ours for the city of Wuhan in China with some differences in both the calibration and the model predictions. Reference [3] analyzed the role of government measures in slowing and reducing COVID-19 growth in different regions in Italy. Reference [4] estimated the contribution of travel restrictions, quarantine, and contact precautions in mitigating the transmission outbreak in Sicily, Italy. Reference [5,6] estimated the evolution of the COVID-19 cases in Wuhan, while [7] investigated the impact of social distancing for the viral spread in the US, and [8] analyzed the impacts of non-pharmaceutical interventions to contain the virus’s expansion in Great Britain. Reference [9] investigated the time evolution of different populations and monitored several parameters for the spread of the disease in China, South Korea, India, Australia, USA, Italy, and the state of Texas in the USA. Reference [10] provides insights on how to estimate the transmission rate of the SIR model when some of the infected people are not identified because they remain asymptomatic.

The main contribution of this paper to the SIR-related literature is the decomposition of the transmission rate between the contagion probability and the number of interpersonal contacts. As our findings show, the differentiation of these two elements is critical for the evaluation of alternative policy interventions aimed at mitigating COVID-19 spread. These results could be crucial for the adequate design of health and economic policies oriented toward containment of the virus.

The paper is organized in two main blocks devoted to describing the methods (Section 1) and discussing the simulation results (Section 2), followed by a review of the main conclusions (Section 3).

## 2. Methods

### 2.1. Model Description

For any given day *t*, we have the decomposition
$$N=x_{t}+z_{t}$$
where *N* is the total population on the arrival day of the first person infected by COVID-19, $x_{t}$ is the accumulated number of people infected by COVID-19 on day *t*, and $z_{t}$ is the accumulated number of people never infected on day *t*. On day 1, $x_{1}=1$ and $z_{1}=N-1.$ For any future day *t*, the law of motion for $x_{t}$ is
(1)$$x_{t}=x_{t-1}+\alpha y\frac{{\tilde{x}}_{t-1}}{N-k_{t-1}}z_{t-1}$$
which adds up to its value on the previous day, $x_{t-1}$, the number of newly infected people $\alpha y\frac{{\tilde{x}}_{t-1}}{N-k_{t-1}}z_{t-1}$. In the latter term, 0<α<1 is the contagion probability on each encounter between one non-infected person and one infected person, y>0 is the number of people each person meets per day, ${\tilde{x}}_{t-1}$ is the number of people currently infected as of day t−1, and $k_{t-1}$ is the accumulated number of deaths caused by COVID-19 as of day t−1.

The ratio $\frac{{\tilde{x}}_{t-1}}{N-k_{t-1}}$ provides the share of currently infected people with respect to the surviving population at the end of day t−1, which determines the probability of meeting someone infected. Thus, the product of the number of encounters by the rate of infected people, $y\frac{{\tilde{x}}_{t-1}}{N-k_{t-1}}$, is the number of infected people every person meets on day *t*. Once we multiply it by the contagion probability on each encounter, we have $\alpha y\frac{{\tilde{x}}_{t-1}}{N-k_{t-1}}$ as the effective daily contagion rate per person. The number of people who have never been infected at the end of day t−1 is $z_{t-1}$, and they are the potential newly infected people (susceptible people in the SIR methodology). Therefore, the second term on the right side of (1), $\alpha y\frac{{\tilde{x}}_{t-1}}{N-k_{t-1}}z_{t-1}$, is the number of newly infected people on day *t*. It explains how the number of new cases depends on both the contagion probability α, and on the total number of encounters between infected and non-infected individuals, $y\frac{{\tilde{x}}_{t-1}}{N-k_{t-1}}z_{t-1}$.

The difference between the accumulated number of people infected, $x_{t}$, and the number of people (still) currently infected, ${\tilde{x}}_{t}$, comes from the fact that the COVID-19 disease is neither chronic nor necessarily lethal. Let us assume, for simplicity, that the outcome of the disease is realized within an interval of days after the incubation period (outcome interval). Thus, if the incubation period of the virus is $T_{i}$ days, where *i* denotes "incubation," the lower bound of the outcome interval is the next day after the end of the incubation period. The upper bound is set to have an outcome interval with the same number of days above and below the average duration of the disease, *T*. Subsequently, the upper bound of the duation of the disease is $T+\left(T-\left(T_{i}+1\right)\right)=2T-\left(T_{i}+1\right)$ days.

The realization of the disease outcome is uniformly distributed along the days of the outcome interval. This is assumed to avoid excessive complexity and due to the uncertainty on the real distribution. Furthermore, there is great case variability in COVID-19 infections due to person-specific characteristics, for example, age, immune system capacity, early diagnosis, and treatment, which makes plausible the assumption of a uniform distribution of the outcome realizations over the days of the outcome interval. Since the number of days with possible realizations of the disease is $2T-\left(T_{i}+1\right)-\left(T_{i}+1\right)+1=2\left(T-\left(T_{i}+1\right)\right)+1$, there is a constant fraction for each daily cohort of infected people, $\frac{1}{2\left(T-\left(T_{i}+1\right)\right)+1}$, who reaches an outcome from the disease on a given day. Therefore, the law of motion for the number of currently infected people by COVID-19 is
(2)$${\tilde{x}}_{t}={\tilde{x}}_{t-1}+\alpha y\frac{{\tilde{x}}_{t-1}}{N-k_{t-1}}z_{t-1}-\left(\frac{1}{2\left(T-\left(T_{i}+1\right)\right)+1}\right)\sum_{j=T_{i}}^{2\left(T-1\right)-T_{i}}\left(x_{t-j}-x_{t-j-1}\right)$$

The individuals of each cohort can either recover (with an associated survival probability 0<1−λ<1) or die (with an associated fatality probability 0<λ<1). The evolution of accumulated deaths, $k_{t}$, is as follows.
$$k_{t}=k_{t-1}+\lambda \left(\frac{1}{2\left(T-\left(T_{i}+1\right)\right)+1}\right)\sum_{j=T_{i}}^{2\left(T-1\right)-T_{i}}\left(x_{t-j}-x_{t-j-1}\right)$$

Naturally, the accumulated number of recovered people, $h_{t}$, is
$$h_{t}=h_{t-1}+\left(1-\lambda \right)\left(\frac{1}{2\left(T-\left(T_{i}+1\right)\right)+1}\right)\sum_{j=T_{i}}^{2\left(T-1\right)-T_{i}}\left(x_{t-j}-x_{t-j-1}\right)$$

Since $N=x_{t}+z_{t}$, we can split up the total infected people in three possible states, $x_{t}=h_{t}+k_{t}+{\tilde{x}}_{t}$, to get
(3)$$N=h_{t}+k_{t}+{\tilde{x}}_{t}+z_{t}$$
which means that total population, *N*, comprises the people who have already healed, $h_{t}$, the people who have already died, $k_{t}$, the people who are infected with their outcome not yet known, ${\tilde{x}}_{t}$, and the people who have never been infected, $z_{t}$.

COVID-19 is an infectious virus that typically causes mild symptoms similar to the common flu, and only a minor fraction of sick people who test positive need hospitalization. In fact, some of the people infected with COVID-19 are asymptomatic, which makes the spreading out of the epidemic more difficult to prevent and control by the health authorities. Reference [11] says that, “Estimates suggest that about 80% of people with COVID-19 have mild or asymptomatic disease”. There is no conclusive evidence on whether the transmission rate from the asymptomatic is different from that of the people who develop symptoms [12]. In the model, we assume that a fraction θ of the infected people who have passed the incubation period, $T_{i}$, suffer from severe complications (typically, respiratory difficulties and pneumonia) and need hospitalization. Thus, the number of hospital beds, $b_{t}$, required to treat COVID-19 positive people on day *t* is
$$b_{t}=\theta \sum_{j=T_{i}}^{2\left(T-1\right)-T_{i}}\left(x_{t-j}-x_{t-j-1}\right)$$
where $\sum_{j=T_{i}}^{2\left(T-1\right)-T_{i}}\left(x_{t-j}-x_{t-j-1}\right)$ is the total number of infected people who have passed the incubation period, $T_{i}$, on day *t*.

To summarize, we have a dynamic system of six equations as follows:$$\begin{array}{ccc} x_{t} & = & x_{t-1}+\alpha y\frac{{\tilde{x}}_{t-1}}{N-k_{t-1}}z_{t-1}\\ {\tilde{x}}_{t} & = & {\tilde{x}}_{t-1}+\alpha y\frac{{\tilde{x}}_{t-1}}{N-k_{t-1}}z_{t-1}-\left(\frac{1}{2\left(T-\left(T_{i}+1\right)\right)+1}\right)\sum_{j=T_{i}}^{2\left(T-1\right)-T_{i}}\left(x_{t-j}-x_{t-j-1}\right)\\ N & = & x_{t}+z_{t}\\ k_{t} & = & k_{t-1}+\lambda \left(\frac{1}{2\left(T-\left(T_{i}+1\right)\right)+1}\right)\sum_{j=T_{i}}^{2\left(T-1\right)-T_{i}}\left(x_{t-j}-x_{t-j-1}\right)\\ h_{t} & = & h_{t-1}+\left(1-\lambda \right)\left(\frac{1}{2\left(T-\left(T_{i}+1\right)\right)+1}\right)\sum_{j=T_{i}}^{2\left(T-1\right)-T_{i}}\left(x_{t-j}-x_{t-j-1}\right)\\ b_{t} & = & \theta \sum_{j=T_{i}}^{2\left(T-1\right)-T_{i}}\left(x_{t-j}-x_{t-j-1}\right) \end{array}$$
which determine the evolution of the 6 endogenous variables $\left\{x_{t},{\tilde{x}}_{t},z_{t},k_{t},h_{t},b_{t}\right\}$, given initial values.

### 2.2. Model Calibration for Spain

The baseline calibration is aimed at representing the outbreak of COVID-19 in a medium-size country. We take the case of Spain because the virus’s spread has been distributed quite evenly within the territory, with a similar evolution in the daily growth of confirmed cases and the reproduction number observed across the Spanish administrative provinces ([13]). As the variables of our model do not incorporate spatial differentiation, we found it suitable for studying the impact of the virus in territories with homogeneous contagion patterns such as Spain (and not other countries that have the pandemic concentrated close to their epicenters, such as China, Italy, and the US).

The total population is N=47 million people to coincide approximately with the population of Spain in 2020. For the fatality rate, λ, we follow [11] who provide an estimated range between 0.3% and 1% with reference on the data released by the World Health Organization. Typically, the infection fatality rate (IFR), defined as $\frac{\mathrm{confirmed}\;\mathrm{deaths}}{\mathrm{confirmed}+\mathrm{unconfirmed}\;\mathrm{cases}}$, is lower than the case fatality rate (CFR), measured as $\frac{\mathrm{confirmed}\;\mathrm{deaths}}{\mathrm{confirmed}\;\mathrm{cases}}$. Since some of the COVID-19 cases are not reported because they are either asymptomatic or the tests have not been taken, these two indicators tend to be quite different, with a higher value of the CFR over the IFR. Our model produces the IFR and assumes that the virus transmission can occur from the following day of contagion. Spain may experience a relatively high IFR due to the population aging (in 2019 people over 75 years old represented 9.54% of the total population) and the much stronger severity of COVID-19 on the elderly. As for capacity, Spain has approximately 300 hospital beds per 100,000 people (below the EU average of about 372 beds), and health coverage is guaranteed by the government with a well-developed public provision of hospitals and treatments. Balancing out these arguments, we set λ=0.0085 (0.85%), above the median value of the range suggested by [11].

The incubation period for COVID-19 is about 5 or 6 days and there is an average period of 10 days or more (longer than a common flu) of confrontation between the immune system and the virus ([11]). Therefore, we set an average disease duration at T=16 days and the incubation period last for 5 days, $T_{i}=5$. Thus, the calibrated outcome interval runs from $\left(T_{i}+1\right)=6$ days after the contagion to $2T-\left(T_{i}+1\right)=26$ days after the contagion.

Reference [8] estimated the COVID-19 hospitalization rate for the population of Great Britain using a subset of cases obtained from China. Their estimate was 4.4%. For Spain, as we assume that in its population there is a higher fraction of elderly people than in either Great Britain or China, we set the hospitalization rate at θ=0.0528 (5.28%), which implies a 20% higher value than the one reported in [8].

The daily number of two-people encounters per day is subject to heterogeneity because it clearly depends on the specific social and economic characteristics of the individuals (the type of job, social/leisure activities, age, etc.), as well as on the social norms and habits of a country or territory. Gatherings for social and economic activities are quite common in Spain. Thus, we set y=25 meetings to represent the average behavior of Spanish citizens in normal times, though recognizing the uncertainty and variance that affect this model parameter. On 14th March 2020, the Spanish government declared a state of emergency, the "state of alarm" (SoA) in response to the COVID-19 outbreak in Spain. The decree contemplated mobility restrictions, school and socioeconomic activity suspensions, and home confinement for the population. Fifteen days after the SoA declaration (29th March 2020), the government passed further actions and enforced home confinement to every person whose job was not related to either health care or basic needs. On 13 April, the government gave legal permission to resume production activity in the manufacturing and construction sectors conditioned on compliance with protective actions to prevent the virus’s spread at the workplace (wearing protection gear, keeping interpersonal distance, reorganizing shifts to minimize workers concentration, etc.). The calibrated model can represent the SoA as a policy intervention that significantly reduces the number of physical contacts among citizens. Hence, we capture the effects of the SoA intervention by reducing, on the SoA declaration day, the number of interpersonal daily encounters from y=25 to y=4. The tighter lockdown actions that came into force 15 days past the SoA declaration are represented as an additional 35% cut in the number of personal contacts to y=2.6 encounters per day. The Spanish Minister of Internal Affairs commented in a press conference on the first day after the suspension of all non-basic economic activities that traffic in public transportation fell 34% compared to the previous working day. Once the tightening was partially relaxed, 30 days past the SoA declaration, the number of daily contacts returned to y=4 and we also cut the contagion probability by 25% to capture the preventive effect of the new working conditions.

The choice of the day in which the isolation is enforced can be crucial for the posterior extension of the disease (as we will document below). Thus, we paid special attention to selecting the day of our model series when the policy intervention took place in Spain. The first confirmed infected person in Spain was a German tourist who tested positive for COVID-19 in La Gomera (Canary Islands) on 29 January 2020. In turn, we consider 29 January as day 1, and the SoA declaration day (14 March) is day 45. The tightening of the SoA, which reduced work permissions only to jobs related to essential needs, took place on 29 March, which is identified as day 60 of the series. The conditioned return to some of the economic activities (13 April) corresponds to day 75.

The contagion probability α measures the speed at which the virus spreads. In Spain, COVID-19 showed exponential growing patterns in the early stages with doubling times for confirmed cases and deaths between 2 and 4 days and a reproduction number, $R_{0}$, between 4.0 and 7.0 ([13]). As the true number of infected people cannot be observed in the data, we have calibrated the value of α to match the series of deaths caused by COVID-19 in Spain (which are comparable between the model simulations and the data). Official data frequently underestimate the number of COVID-19 deaths because many casualties take place outside hospitals (homes, elderly residences). Reference [14] found these missing deaths to be a very large number for the outbreak in Spain by comparing the excess over the historical average of mortality registration with the official number reported by the government. Specifically, Reference [14] calculated that by 5 April 2020, the number of accumulated deaths caused by COVID-19 in Spain should be 19,700 instead of the officially reported value of 12,400 (7300 missing deaths). We have chosen the value of the primary contagion probability, α, to match the datapoint of 19,700 deaths on 5 April in the series of accumulated deaths generated by the model. This criterion determined setting α=0.01615. Figure 1 shows the official data and model simulations of accumulated and daily deaths caused by COVID-19 in Spain. Both the phases and peak day of the curve of daily deaths are well replicated by the model, with the gap due to the missing deaths reported by [14]. Such a difference tends to shrink over the downward phase, which is consistent with a larger number of tests taken and the mitigation of the problems for the diagnostic provisions that have characterized the peak days of the COVID-19 epidemic in Spain.

## 3. Simulation Results

We have programmed the simulations in Matlab. For initial values, we considered that on day 1, t=1, there was one imported contagion and one person got infected while the rest of the population had no virus; i.e., $x_{1}={\tilde{x}}_{1}=1$. Then, we ran the calibrated six-equation model forward over the next 365 days to analyze the effects of the SoA declaration for the COVID-19 spread in Spain. In addition, we simulated the model under alternative decisions on the timing and intensity of the policy intervention. The variables to be discussed here are the number of infected people, accumulated deaths, and the number of hospital beds required to treat COVID-19 (infected people who need hospitalization). The benchmark case is the "no intervention" scenario, keeping y=25 as calibrated for normal times in Spain. If there would have been no intervention, the model prediction is that almost all the Spanish people would have been infected (46.95 million people), and applying the fatality rate, 0.85% of them (nearly 400 thousand people) would have died.

The estimated effects of the SoA intervention are displayed as red lines in Figure 2. In comparison to the no intervention scenario (black lines), the curves of infected people and hospitalized people shift down and widen up as a clear example of the "flattening of the curve" pattern. If we compare the numerical values (reported in Table A1 of the online Appendix A), we find impressive effects. Thus, the SoA declaration is estimated to have reduced the accumulated number of infected people from 46.95 million to 5 million, the maximum number of people who needed/will need hospitalization from nearly 2.4 million people to 155 thousand people (a 93.5% cut), and accumulated deaths from almost 400 thousand to 42.5 thousand (a 89% cut).

### 3.1. The Timing of Social Distancing

As shown in the bottom right-hand cell of Figure 2, the number of people who need to be hospitalized shows no apparent variation in the first days after the SoA declaration in comparison to the no intervention case. The reason for this lack of effect is that the reduction in the hospitalized people will not be realized before the end of the 5-day incubation period. Precisely, it is day 51 (6 days after the SoA day) when the slope of the red line flattens, as there are fewer infected people who develop symptoms and need to be hospitalized. We represent the Spanish hospital bed capacity as the horizontal dashed line in the diagram of the bottom right-hand side of Figure 2 and Figure 3. Spain has an overall amount of around 141,000 hospital beds. Let us suppose that in normal times the capacity utilization rate is 60%. Thus, the hospital beds’ capacity to cope with the COVID-19 spread in Spain is assumed to be 60% of 141,000, which is 84,600 units. The model estimates that between days 50 and 78 (nearly one month) the demand for hospital beds exceeds capacity. The downward phase is fast for some days after the peak day but it turns slower on day 70 onwards (coinciding with the end of the outcome interval assumed in the calibration).

A 4-day earlier intervention (day 41) would have prevented many infections and reduced the number of deaths and the hospitalization needs (see the flattening and pushing down of the green lines in Figure 2 relative to the red lines and numbers reported in Table A1 of the Appendix A). In a scenario with social distancing enforced 4 days earlier, the model estimates a reduction by 67% in the accumulated numbers of infected people (from 5 million to 1.65 million) and deaths (it was estimated that 28.5 thousand lives would have been saved). Moreover, the number of required hospitalizations drops by 71% on peak day, from 155,100 to 44,300, which could have been totally covered by the Spanish health care system.

The 4-day postponement of the intervention to day 49 would have increased infected people, hospitalization needs, and deaths by a factor close to 2.5 (see the blue lines in Figure 2). The situation would have been catastrophic for the more than 330 thousand people who would have needed medical treatment on the peak day—this number is more than six times the Spanish hospitalization capacity.

In short, the simulation results indicate that the choice of the day for setting the enforcement of social distancing has critical consequences on the evolution of the virus’s spread.

### 3.2. The Intensity of Social Distancing

The effects of different degrees of intensity of the social distancing action taken by the Spanish government are displayed in Figure 3, with some numbers documented in Table A2 of the Appendix A. Thus, we compare the cases of y=3 (more intensity on isolation) and y=5 (less intensity on isolation) to the calibrated setting of y=4 for the SoA procurement. Once again, the quantitative effects are very large (although somehow not as large as they were for the timing of the intervention).

The green lines on Figure 3 indicate that only reducing the SoA enforcement in one more interpersonal meeting would produce an estimated decrease in the number of accumulated deaths by 34% (from 42.5 thousand to 28 thousand) and in the peak number of people who need hospitalization by 19% (from 155 thousand to 126 thousand). By contrast, a looser implementation of the SoA with y=5 daily encounters per person would have an important cost in human lives (the accumulated number of deaths would rise by 30 thousand) and on the number of people who need hospitalization (on peak day 37 thousand more). Actually, the health care system would be on the verge of collapsing because for 45 consecutive days (between day 50 and day 94, both included) more hospital beds would be required than the installed capacity.

### 3.3. The Effects of Isolation Enforcement on the Epidemic Duration

Next, we analyze the duration of the epidemic under alternative scenarios. Usually, social distancing and isolation policies are considered to cause the flattening of the curve on the epidemic characterized by both lower peak values (shift down and widening of the curve) and a later observation of these peak values (shift to the right of the curve). The delay on the observed peak is sometimes used by commentators and policy makers as a justification for not implementing a severe isolation enforcement due to a longer epidemic duration. The daily series of “currently infected people” in Figure 2 and Figure 3 show that the peak day is observed around day 60 in all cases of policy intervention. In the no isolation enforcement case, the peak day occurs on day 60.

Since Figure 2 and Figure 3 are truncated from above, the online Appendix A includes a Figure with a full-sized vision of the series of currently infected people. This illustrates the dramatic effects of isolation to produce the flattening of the curve. This full-sized Figure also shows that the isolation policies do not involve any shifting of the curve to the right because the peak day is not delayed following any isolation intervention.

The duration of the epidemic is apparently similar in all active cases displayed in Figure 2 and Figure 3, as by day 100 numbers converge towards the zero line on the number of currently infected people. From the SoA declaration, day 45, to approximately day 85, the number of currently infected people without intervention is much higher than any case of isolation enforcement. After day 85 or so, Figure 2 and Figure 3 seem to indicate that the black line (no intervention) falls below the other lines (isolation enforcement).

Although peak days are anticipated due to isolation, the downsizing of the epidemic is faster under the no intervention than with any case of isolation enforcement. In quantitative terms, the model simulations indicate that on day 105 (60 days after the SoA declaration), the no intervention scenario would have 1000 infected people (0.003% of the value on peak day) while the number under the SoA enforcement would still be 254 thousand (13% of the value on peak day). Either earlier or stricter isolation actions reduce the number of infected people to 92 thousand or 65 thousand, respectively. If we look ahead at 90 days after the SoA declaration, all scenarios would lead to small numbers of remaining infected people (virtually 0 for the no intervention case and between 9 thousand and 71 thousand with isolation enforcement). These numbers (reported in Table A3 available in the online Appendix A) call for a cautious design of the calendar for the isolation downsizing that gradually restores the economic and social activities when the number of active cases is sufficiently low.

### 3.4. A Second Peak?

The state of alarm in Spain contemplated a gradual return to normality from 11 May (day 102), when the general quarantine period ended and many of the isolation enforcement actions (home lockdowns and mobility restrictions) ceased. Family gatherings were permitted with some limitations on the duration and the number of relatives involved. In addition, most shops, bars, restaurants, and hotels reopened, subject to controls on interpersonal distance and continuous disinfection. In turn, the number of daily encounters between Spanish people increased. For containment, health authorities announced that some activities would remain suspended (schools, music concerts, certain sports competitions, etc.), preventive actions would be required for both working and using public transportation (wearing face masks, regular disinfections and hand washing, keeping a 1.5-m interpersonal distance), and substantial public provisions of tests for rapid identification and self-isolation of positive cases were available. These mitigation actions were designed to cut down the contagion probability. The after-lockdown stage of the epidemic is therefore characterized by a higher value in the number of daily meetings per person, *y*, and a lower value in the primary contagion probability, α. These two changes have opposing effects on the contagion pace that have not been considered so far and will be discussed next. Since αy is the product of the primary contagion probability, α, times the number of daily encounters, *y*, we can refer to it as the maximum contagion probability (i.e., the one associated with the case of meeting infected people in all the daily encounters). The calibrated value of αy for the SoA stage prior to the end of lockdown is αy=(0.75)(0.01615)(25−21)=(0.0121)(4)=0.0484 (4.84%). Taking αy=4.84% as the benchmark value, we will examine the evolution of the COVID-19 curve under three possible scenarios for αy with the reopening of socioeconomic activities:-A high value of maximum contagion probability: αy=(0.01)(10)=0.10 (or 10%).-A moderate value of maximum contagion probability: αy=(0.01)(8)=0.08 (or 8%).-A low value of maximum contagion probability: αy=(0.01)(6)=0.06 (or 6%).

These scenarios combine a lower primary contagion probability (α falls from 0.0121 to 0.01) with a higher number of daily encounters per person (*y* rises from 4 to 10, 8, or 6). Figure 4 shows the results.

Small changes in the value of the maximum contagion probability αy result in quite different trajectories for the COVID-19 spread in Spain. When αy rises from 4.84% to 10%, the curve of currently infected people quickly bends uphill with a 100-day long period of continuous increase (see the blue line in Figure 4). Hence, the resulting second wave would be even worse than the one suffered in March: it would last longer and the peak number of infected people would be 2.85 million. The effects in the accumulated number of infected people and deaths would be dramatic (see numbers reported in Table A4 of the Appendix A).

A more moderate increase in the maximum contagion probability from 4.84% to 8% would still produce a second peak of the virus’s spread but with a smaller prevalence than the first peak. As the red line of Figure 4 shows, the number of infected people would feature a low positive slope from May to September. On the second peak day (around mid-September), the number of currently infected people is 462 thousand, approximately 1/4 of the value observed in late March. The death toll and the accumulated number of infections would be more than double the numbers obtained with no mitigation of social distancing because the COVID-19 epidemic would be present in Spain for the whole year.

If the increase of the maximum contagion probability after 11 May were to be small (from 4.84% to 6%), there would be no second wave of the COVID-19 outbreak and the curve would keep moving downhill (see green line of Figure 4). The accumulated numbers of infected people and death would barely increase. This result shows that a second wave can be avoided if the change in the maximum contagion probability, αy, is sufficiently low. We have searched for the threshold of αy that determines whether the curve turns upward or continues downward. Such a critical level is found at αy=0.0761 (7.61%). Thus, the Spanish health authorities should monitor that the maximum contagion probability stays below 7.61% to prevent a second COVID-19 peak.

## 4. Conclusions

What impact does the timing and intensity of social distancing have on flattening the COVID-19 curve? We presented a dynamic discrete-time model of the COVID-19 spread that provides information on six variables relevant for the quantitative analysis to answer this question.

The model has been calibrated to Spanish data to quantify the impacts of alternative isolation enforcement means in the face of the COVID-19 pandemic. Compared to the no intervention scenario, the state of alarm declaration is estimated to cut the number of accumulated deaths by 89% and the number of hospital beds needed by 93.5%. Both an earlier and a more intense intervention could have been crucial for further reductions in infected people, deaths, and hospitalizations. The isolation enforcement does not delay the peak day of the epidemic but slows down its end.

The model estimates that on the day of the state of alarm declaration (14 March) in Spain, there were 1.5 million actively infected people; at peak day (27 March) that number reached 1.9 million; and on the last day of forced home confinement (10 May) it dropped to 300 thousand. The mitigation of isolation enforcement could bring a second wave of the COVID-19 outbreak in Spain if the maximum contagion probability rises from 4.84% to beyond 7.61%.

## Figures and Tables

**Figure 1 ijerph-17-07283-f001:**
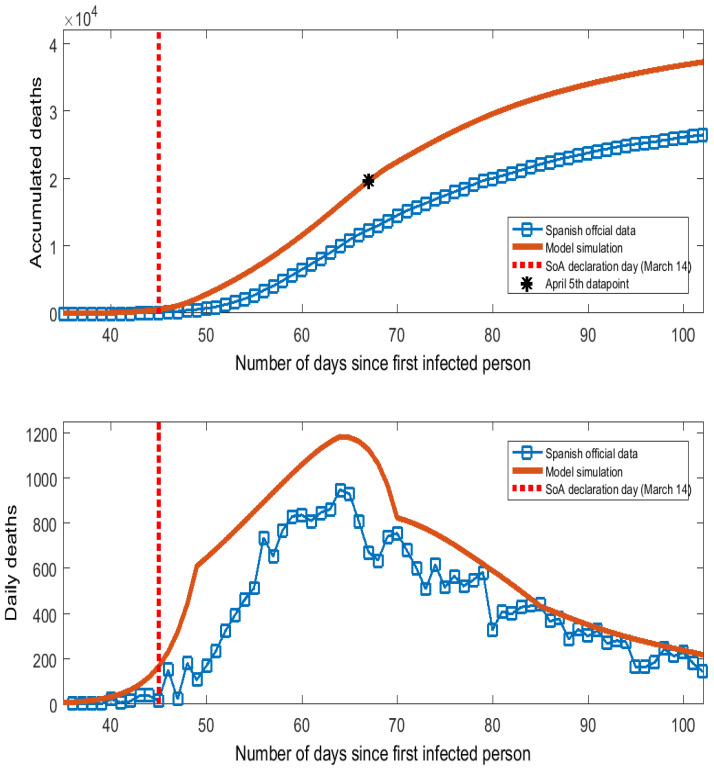
Deaths caused by COVID-19 in Spain.

**Figure 2 ijerph-17-07283-f002:**
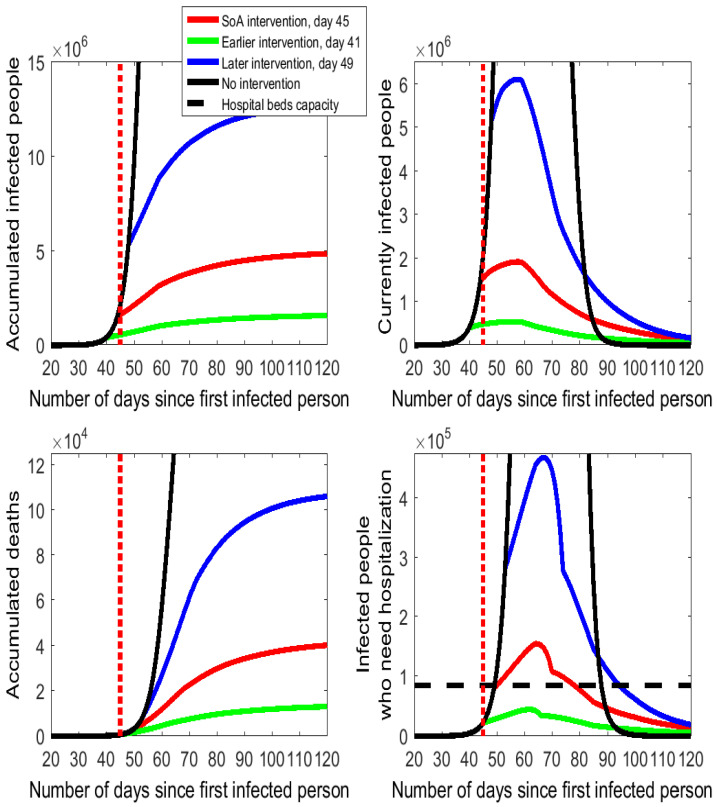
Alternative timings for the isolation policy in Spain following the COVID-19 outbreak.

**Figure 3 ijerph-17-07283-f003:**
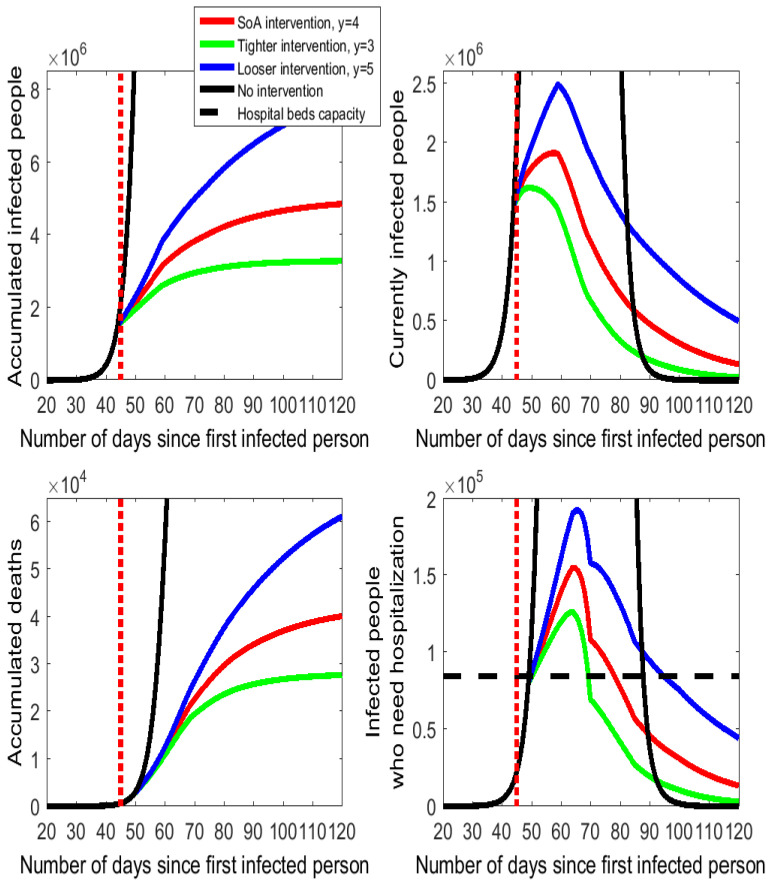
Alternative intensities for the isolation policy in Spain following the COVID-19 outbreak.

**Figure 4 ijerph-17-07283-f004:**
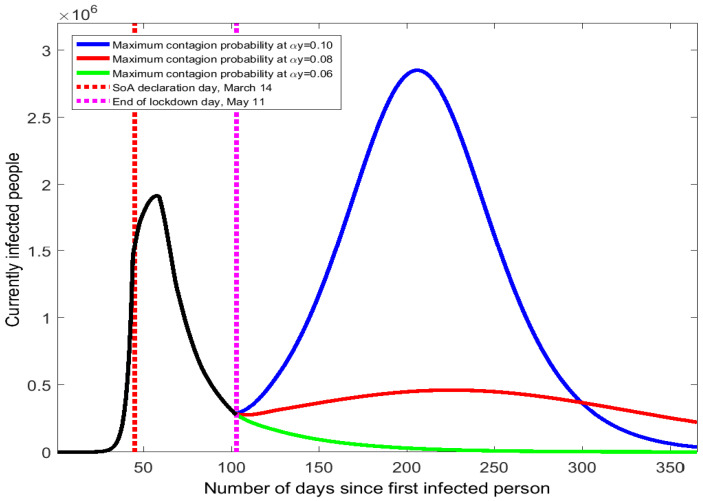
COVID-19 contagion spread Spain after the end of the state of alarm.

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
