# Peer review of "The Timing and Intensity of Social Distancing to Flatten the COVID-19 Curve: The Case of Spain"

_ijerph, 2020, doi:10.3390/ijerph17197283_

Round 1

Reviewer 1 Report

Data obtained from the literature mentioned in the Introduction comprise a well documented theoretical background, enabling determination of the study aim and correct choice of its realization methods.
It should be considered whether points: 1. Model description and 2. Model calibration for Spain should constitute a chapter of the Methods.
The authors presented their simulation results systematically and illustrated them with graphs/tables that are helpful in the text reading.
I value highly the Authors’ ability for a clear summary and data interpretation.
In summary, I can state that the aim of the study was achieved.
The presented studies have a significant epidemiological value.

Author Response

See attached response letter

Reviewer 2 Report

Dear Authors,

I appreciated your manuscript.

It is original, in the field of COVID pandemic, and it sounds as scientific contribute. 

The global quality of presentation is good and the interest for the readers could be high.

I consider the abstract informative.

For what concern the introduction I suggest you to add more references.

I consider the paper ready for publication afeter minor revisions (in itroduction section)

Author Response

See attached response letter

Reviewer 3 Report

I found this manuscript by Casares and colleagues interesting and well presented. The introduction provides appropriate references even if I would suggest to discuss on previous studies instead of provide a mere list. 

Methods are adequately described but I would suggest to stress that this model is based on a set of assumptions that must be recognize here and in the discussion section. Indeed, assumptions - such as those referred to incubation period, fatality rate etc - might signifcantly affect both model calibration and estimates. Accordingly, I would suggest the Authors to perform a sensitivity analysis in order to evaluate if the model is sensitive to specific parameters.

Moreover, I would kindly ask how the author handled undocumented infections. Indeed, it has been established that in other contries there was a significant proportion of positive patients not documented (please consider doi: 10.3390/jcm9051350).

With respect to figures, I noted an overlap between images and figure titles, please check it. 

In the discussion section, i would suggest to better explain the main limitations of this study

Author Response

See attached response letter

Round 2

Reviewer 3 Report

None